# Multidimensional Frailty and Vaccinations in Older People: A Cross-Sectional Study

**DOI:** 10.3390/vaccines10040555

**Published:** 2022-04-03

**Authors:** Nicola Veronese, Giusy Vassallo, Maria Armata, Laura Cilona, Salvatore Casalicchio, Roberta Masnata, Claudio Costantino, Francesco Vitale, Giovanni Maurizio Giammanco, Stefania Maggi, Shaun Sabico, Nasser M. Al-Daghri, Ligia J. Dominguez, Mario Barbagallo

**Affiliations:** 1Geriatric Unit, Department of Health Promotion, Mother and Child Care, Internal Medicine and Medical Specialties “G. D’Alessandro”, University of Palermo, 90127 Palermo, Italy; giusy.vassallo@community.unipa.it (G.V.); maria.armata@community.unipa.it (M.A.); laura.cilona@community.unipa.it (L.C.); salvatore.casalicchio@community.unipa.it (S.C.); roberta.masnata@community.unipa.it (R.M.); ligia.dominguez@unipa.it (L.J.D.); mario.barbagallo@unipa.it (M.B.); 2Hygiene Section, Department of Health Promotion, Mother and Child Care, Internal Medicine and Medical Specialties “G. D’Alessandro”, University of Palermo, 90127 Palermo, Italy; claudio.costantino01@unipa.it (C.C.); francesco.vitale@unipa.it (F.V.); 3Microbiology Section, Department of Health Promotion, Mother and Child Care, Internal Medicine and Medical Specialties “G. D’Alessandro”, University of Palermo, 90127 Palermo, Italy; giovanni.giammanco@unipa.it; 4Consiglio Nazionale delle Ricerche, Neuroscience Institute, 35128 Padova, Italy; stefania.maggi@in.cnr.it; 5Chair for Biomarkers of Chronic Diseases, Biochemistry Department, College of Science, King Saud University, Riyadh 11451, Saudi Arabia; ssabico@ksu.edu.sa (S.S.); ndaghri@ksu.edu.sa (N.M.A.-D.); 6Faculty of Medicine and Surgery, University of Enna “Kore”, 94100 Enna, Italy

**Keywords:** multidimensional prognostic index, vaccination, influenza, herpes zoster, pneumococcus, diphtheria-tetanus-pertussis

## Abstract

It is known that influenza, herpes zoster, pneumococcal and pertussis infections may increase morbidity and mortality in older people. Vaccinations against these pathogens are effective in older adults. Frailty seems to be an important determinant of vaccination rates, yet data supporting this association are still missing. Therefore, we aimed to investigate the prevalence of four recommended vaccinations (influenza, herpes zoster, pneumococcal and diphtheria-tetanus-pertussis) and the association with multidimensional frailty assessed using a self-reported comprehensive geriatric assessment tool, i.e., the multidimensional prognostic index (SELFY-MPI). Older participants visiting the outpatient clinic of Azienda Ospedaliera Universitaria, Palermo, Italy were included. The SELFY-MPI questionnaire score was calculated based on eight different domains, while the vaccination status was determined using self-reported information. We included 319 participants from the 500 initially considered (63.8%). Vaccination against influenza was observed in 70.5% of the cases, whilst only 1.3% received the vaccination against diphtheria-tetanus-pertussis. Participants with higher SELFY-MPI scores were more likely to report vaccination against pneumococcus (45.6 vs. 28.3%, *p* = 0.01), whilst no significant differences were observed for the other vaccinations. In conclusion, the coverage of recommended vaccinations is low. Higher SELFY-MPI scores and vaccination status, particularly anti-pneumococcus, appear to be associated, but future studies are urgently needed for confirming that frailty is associated with vaccination status in older people.

## 1. Introduction

Infectious diseases (IDs) are among the most important causes of hospitalization and mortality in older people [1]. It is widely known that influenza, herpes zoster, pneumococcal and pertussis infections may increase the morbidity and mortality rates in older people [2]. However, all these IDs are largely preventable through vaccination.

The vaccines protecting against these well-known diseases have been available for a long time, even if their efficacy and persistence of immunity are often suboptimal, especially in very old individuals [2,3]. This is probably due to the different characteristics of older people in which frailty is often present [4,5,6]. Frailty, a condition characterized by a reduced reserve against stressors [7], is an important determinant of the lower efficacy of vaccinations in older populations. Immunosenescence (i.e., the progressive decline in immunity with age, affecting both innate and adaptive immunity) is indeed more common in frail than in robust older people [8]. In this regard, a close interaction between frailty, often due to the presence of some comorbidities (such as dementia, sarcopenia and infectious diseases), and immune function has been identified in older adults, clearly showing that a well-functioning immune system may prevent frailty and its consequences and that a strict adherence to an immunization schedule not only may delay frailty, but also maintain the immune homeostasis [9].

However, epidemiological data regarding the vaccination status in older individuals and its association with frailty has been poorly explored until now. Given this background, we aimed to investigate the coverage of four recommended vaccinations (influenza, herpes zoster, pneumococcal and diphtheria-tetanus-pertussis) in a cohort of older people and the association with multidimensional frailty assessed using a self-reported comprehensive geriatric assessment tool, i.e., the multidimensional prognostic index (SELFY-MPI).

## 2. Materials and Methods

### 2.1. Data Source and Subjects

All participants of both sexes, aged 60 years and above, accessing the outpatient clinic of the Azienda Ospedaliera Universitaria “Paolo Giaccone”, Palermo were initially contacted during the administration of the SARS-CoV-2 vaccination. Only those who refused to sign the informed consent were excluded. The presence of a caregiver was allowed to eventually assist the participants in filling out the SELFY-MPI questionnaire and vaccination status. Possible eligible participants were contacted during the administration of the SARS-CoV-2 vaccination by physicians in training in geriatric medicine of the University of Palermo, which explained the main aims of the work to the possible participants and their caregivers.

The study was approved by our local ethical committee on 19th May 2021 (protocol no. 5) and the study lasted six months, from 1st June 2021 to 31st December 2021.

### 2.2. The Multidimensional Prognostic Index (SELFY-MPI)

For the aims of this work, we used the self-administered version of the MPI, i.e., the SELFY-MPI [10,11,12], which considered the following domains:The functional status was assessed through the Barthel activities of daily living (ADL) [13] scale that includes abilities in feeding, bathing, personal hygiene, dressing, fecal and urinary continence and toilet use.The Barthel Mobility scale [13], which includes transfer from bed to chair or wheelchair, walking and going up and down the stairs.Independence by means of the Lawton’s IADL (Instrumental ADL) scale [14], as reported earlier.Cognitive status assessed through the self-administered cognitive screening test (test your memory) [15]. It is a validated 10-task cognitive test exploring several domains, including memory, semantic knowledge and visuospatial skills. The score ranges from 0 to 50, higher scores indicating better cognitive function [16].Nutritional status investigated with the Mini-Nutritional Assessment-Short Form (MNA-SF) [17], as reported in the previous section. A validated self-administered MNA-SF was used [18].Number of medications.Comorbidity: CIRS (Cumulative Illness Rating Scale) comorbidity index is the number of health problems/diseases that are so severe as to require chronic drug therapies in 13 aspects of health [19]. CIRS can be consequently self-assessed by reporting health problems/diseases that require medications for their treatment.Social aspects, categorized as living alone, with family/formal caregiver and in a nursing home.

The sum of the calculated scores from the eight domains was divided by the number of domains to obtain a final risk score ranging from 0 = no risk to 1, higher values indicating a higher risk of mortality [20]. MPI is commonly used as a tool for evaluating the presence of frailty, assessed through a comprehensive geriatric assessment, i.e., multidimensional frailty [21]. For the aims of this work, we used as cut-offs 0.25 and 0.38 for creating tertiles (SELFY-MPI 1, MPI 2 and MPI 3). We also reported descriptively the prevalence of multidimensional frailty (MPI > 0.66) and prefrailty (MPI between 0.33 and 0.66), according to the preestablished cut-offs [20]. The median time required to complete the MPI is about 16 min [20].

### 2.3. Questions Regarding Vaccination Status

The vaccination status was explored using self-administered information with anonymous data through four specific questions: (1) Have you been vaccinated against the flu this year or last year? (2) Have you been vaccinated against pneumococcus in the last five years? (3) Have you been vaccinated against shingles during your life? (4) Have you been vaccinated against diphtheria, tetanus and pertussis (whooping cough) in the last five years? For each question, three answers (yes, no and don’t remember/don’t know) were possible. Vaccination recommendations in individuals aged over 60 years in Italy were better detailed in Reference [22].

### 2.4. Statistical Analysis and Sample Size Calculation

The sample size was calculated according to a work investigating the prevalence of multidimensional frailty with the SELFY-MPI among general practitioners in Italy [23]. In this regard, hypothesizing a prevalence of multidimensional frailty of 3.99%, a type I error of 5% and a power of 80%, 246 participants should be enrolled. Unfortunately, no other works, to the best of our knowledge, were available regarding the frailty status and vaccination coverage in older people.

Continuous variables were normally distributed according to the Kolmogorov–Smirnov test for age and for SELF-MPI scores and its domains (i.e., mobility, ADL, IADL, MNA, CIRS and number of medications). Therefore, data were shown as the means and standard deviation values (SD) for quantitative measures. Percentages were used for discrete variables, such as the percentage of females and those living alone. Levene’s test was used to test the homoscedasticity of the variances, and if its assumption was violated, Welch’s ANOVA was used. *p*-values were calculated using the Jonckheere–Terpstra test [24] for continuous variables and the Mantel–Haenszel chi-square test for categorical variables by SELFY-MPI tertiles. All analyses were performed using SPSS 20.0 for Windows (SPSS Inc., Chicago, IL, USA). All statistical tests were two-tailed, and statistical significance was assumed for a *p*-value < 0.05.

## 3. Results

Overall, a total of 500 participants were initially contacted, and 319 participants (response rate = 63.8%; mean age: 77.5 ± 7.6 years; range: 60–97), mainly females (58.0%), were included in this cross-sectional study. The mean SELFY-MPI was 0.33 ± 0.16, with a range between 0.06 and 0.75. The assistance of a caregiver was necessary for 207 participants, i.e., 64.9% of the participants included.

Table 1 shows the descriptive characteristics of the participants divided by their MPI values in tertiles. As expected, participants in the SELFY-MPI 3 category (MPI > 0.38), indicating frailer individuals, were significantly older and more likely to be females than their counterparts. Participants in the SELFY-MPI 3 group scored significantly worse in all MPI domains (ADL, mobility, nutritional and cognitive status, number and severity of medical conditions and number of medications) than participants in SELFY-MPI 1 and SELFY-MPI 2, except in IADL (*p* = 0.28). Finally, people in SELFY-MPI 3 had the greatest prevalence of living alone than their counterparts (31.5% vs. 7.9% in the SELFY-MPI 1 group).

As shown in Appendix A, the high prevalence of severe comorbidities and cognitive impairments were more likely to contribute to multidimensional frailty, since 63.0% had high risk values in the CIRS-SI domain and 25.4% cognitive difficulties. Using the classical division of MPI, 41.1% could be considered prefrail (MPI between 0.33 and 0.66) and 3.4% frail (MPI > 0.66). Regarding the number of vaccinations, 84 participants (26.3%) reported that they did not receive any of the four vaccinations investigated, 116 (36.4%) only one, 107 (33.5%) two and only nine participants (2.8%) three vaccinations. No participant received all four vaccinations. Overall, vaccination against influenza was the most common vaccine received in 70.5% of the cases, whilst only 1.3% received the vaccination against diphtheria-tetanus-pertussis (Table 2). Regarding the last vaccination, 15.9% were unable to recall the status of their vaccination.

Figure 1 reports the association between the SELFY-MPI and frequency of each of the four vaccinations in the participants included. People in SELFY-MPI 3 reported more frequently the vaccination against pneumococcus compared to participants in SELFY-MPI 1 (45.6 vs. 28.3%, *p* = 0.01), whilst no significant differences were observed for influenza (*p* = 0.29), herpes zoster (*p* = 0.71) or diphtheria-tetanus-pertussis vaccination (*p* = 0.62). Finally, people in SELFY-MPI 3 reported a significantly higher mean number of vaccinations compared to people in SELFY-MPI 1 (0.97 ± 0.80 vs. 1.19 ± 0.81, *p* = 0.046).

## 4. Discussion

In this cross-sectional study, we reported that the vaccination coverage recommended for older individuals by the Italian National Vaccination Plan and by most of the international agencies, such as the World Health Organization (WHO) and ECDC (European Centre for Disease Prevention and Control), is generally low. However, frailer people reported a significantly higher number of vaccinations and a higher vaccination coverage of anti-pneumococcal vaccination than their robust counterparts.

The first epidemiological aspect that we would like to discuss is the vaccination coverage observed in our cohort. This figure, in fact, ranged from 70.5% of the participants in the case of influenza vs. only 1.3% against diphtheria-tetanus-pertussis, with a small part receiving three vaccinations over four suggested (2.8%) and no one receiving all four. Regarding influenza, our data are in line with other epidemiological observations all over the world [25]. In 2003, countries participating in the World Health Assembly established that a minimum coverage against influenza of at least 75% should be reached by 2010, with the optimum coverage rate being set at 95% [25]. Finally, the recent COVID-19 epidemic has raised the adherence to influenza vaccination also in older people [26,27]. On the contrary, the data regarding the other vaccinations investigated in our study are still low. For example, the vaccination coverage of the anti-pneumococcus vaccination was 37.6%, even if this vaccination is recommended in Italy to all people aged more than 65 years [28]. In this regard, it was previously reported that substantial differences in the recommendations exist across various national immunization technical advisory groups in the world [29], despite anti-pneumococcus vaccination being highly effective in reducing hospitalization and the mortality rate in older people [30]. In this regard, it is important to remember that not only the prevalence of an antibiotic resistant to *S. pneumoniae* has increased over the last years but that older age is an important determinant of pneumococcal mortality [31].

Furthermore, we reported that only 2.5% of the population included received the vaccination against herpes zoster. This finding is somewhat surprising, since increasing epidemiological data have reported that herpes zoster is a condition typical of older adults, with severe potential sequelae, and we have the availability of safe and effective vaccines [32,33]. Older people, in fact, have an incidence rate of herpes zoster more than ten times higher than younger people [32,33], and vaccination is highly effective in preventing not only the disease but also the complications associated, such as post-herpetic neuralgia [34]. Again, it is key to remember that the incidence of herpes zoster is dramatically higher in older people, and it is often associated with solid and hematological tumors, typical of older people [31].

Finally, the lowest vaccination coverage among the four vaccinations was observed for diphtheria-tetanus-pertussis. Even if the vaccination coverage in children against diphtheria-tetanus-pertussis is overall good [35], only one adult in three is protected against these infections [36], and this figure is probably lower in older adults [37]. Our study reported a significantly lower vaccination coverage of vaccination against diphtheria-tetanus-pertussis when compared to US data [37]. In Italy, a lower public awareness for these conditions in the aged population and inadequate access to this specific vaccination could probably explain our findings [38]. In this regard, it is mandatory to remember that 82% of the tetanus cases between 2001 and 2010 occurred in older people [39]. Moreover, registries that could allow reminder/recall notes for the booster doses are not available, and people are unaware of the need to receive the booster. As shown in our study, 15.9% of participants did not know their vaccination status.

Another important finding is the potential association between vaccination status and frailty, assessed using a self-reported multidimensional assessment. Increasing literature is reporting that the presence of frailty should be assessed using a comprehensive geriatric assessment and, so, with a multidimensional tool [21,40]. Moreover, to the best of our knowledge, SELFY-MPI is the only tool for self-reporting frailty available that cannot be considered only as a screening tool. As reported in other works, in older people, frailty, assessed using a multidimensional approach, has a prevalence of about 5%, and more importantly, prefrailty may affect about one person out of two [6,23,41]. In our study, we found a slightly lower prevalence of frailty and prefrailty, according to a multidimensional approach, probably because these older people were able to access the services for receiving vaccination against SARS-CoV-2; therefore, bedridden and more disabled, older participants were not included. The potential association between vaccinations and frailty was extensively assessed by previous literature, giving a paradoxical picture: frail participants who need vaccines for fighting IDs are also the ones more susceptible to adverse reactions [9]. At the same time, strong evidence of a possible association between IDs with disability and cognitive impairments, two domains strictly associated with frailty, exists [9]. Finally, another possible novel finding of our study is that frailer people were more likely than their counterparts to be vaccinated against pneumococcus. We can hypothesize that general practitioners may have advised these participants more frequently to receive the vaccination against pneumococcus than their more robust counterparts for fear of pneumonia [42] which is unfortunately extremely high in frail older people [42].

However, from an epidemiological point of view, no other studies reported the vaccination coverage of the recommended vaccinations by frailty status, even if these data could be important for better tailoring this preventative action in frail older people. Frailty is associated with an impaired immune response in older ages, likely due to immunosenescence, indicating the need for stronger actions directed at frail older people in terms of the access and delivery of vaccinations, besides the already existing vaccination strategies for older adults, such as the use of adjuvanted or high-dose vaccines. Our study overall suggests that frailer people, indicated as higher SELFY-MPI values, reported a higher number of vaccinations and a higher vaccination coverage against pneumococcus, even if more than half of the participants in this subgroup were not vaccinated against this pathogen. These findings seem to suggest that physicians recommend this vaccination in frail patients, often affected by comorbidities, as a secondary prevention measure to avoid the exacerbation of preexisting conditions associated with pneumonia. Unfortunately, in this scenario, we miss an opportunity to promote healthy aging in the most robust ones, because it is clear that pneumonia is associated with negative outcomes, such as hospitalization and CVD complications, in the older population.

On the contrary, no significant differences emerged for the other vaccinations investigated. To the best of our knowledge, this is the first research to report epidemiological data regarding this possible association, and future studies are urgently needed.

The findings of our study must be interpreted within its limitations. First, the cross-sectional nature does not permit understanding any causal association between vaccination status and frailty. Second, since self-reported information regarding frailty and vaccination status was used, a recall bias could be present, as well as the caregiver assistance that was necessary for several participants. However, we believe that, for epidemiological purposes, the use of self-reported tools should be encouraged, since it is important for evaluating the perceived needs and health status of older participants [43], whilst the use of a questionnaire to assess the vaccination coverage can introduce a recall bias. Finally, the results of the present study are locally or regionally applicable; therefore, the external validity (generalizability) of these findings must be verified in larger studies conducted in other populations.

## 5. Conclusions

In this study on community-dwelling older people undergoing vaccination against SARS-CoV-2, the vaccination coverage of recommended vaccinations was low. Our study suggests that higher SELFY-MPI scores, indicating more propensity toward a multidimensional frailty, and vaccination status, particularly anti-pneumococcus, appear to be associated. Future studies, particularly with a longitudinal design, are urgently needed to confirm our findings. Our study underlines again the important problem of considering frailty in terms of a multidimensional approach in older people and, also, for suggesting vaccinations that can prevent the transition from frailty to disability.

## Figures and Tables

**Figure 1 vaccines-10-00555-f001:**
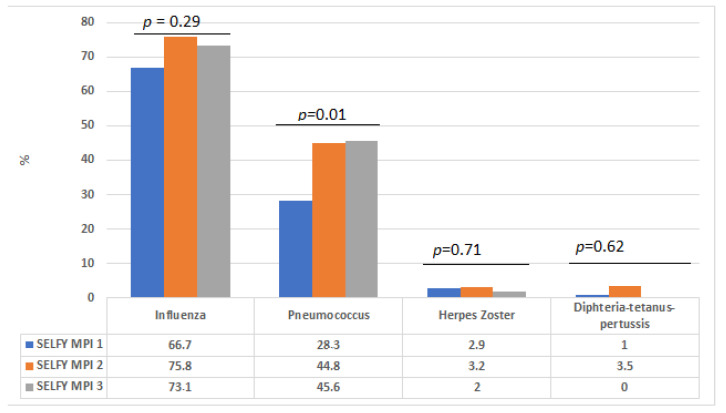
Vaccination coverage by SELFY-multidimensional prognostic index values.

**Table 1 vaccines-10-00555-t001:** Descriptive characteristics of the sample included by SELFY multidimensional prognostic index values.

Parameters	SELFY-MPI ≤ 0.25 (*n* = 112)	SELFY-MPI 0.26–0.38 (*n* = 101)	SELFY-MPI > 0.38 (*n* = 106)	*p*-Value
Age	75.9 (7.6)	77.3 (7.6)	79.5 (7.2)	0.002
Females (%)	50.9	51.5	71.7	0.002
ADL	1.4 (2.4)	3.5 (6.4)	7.3(11.6)	<0.001
IADL	5.7 (2.1)	5.3 (2.6)	5.2 (3.0)	0.28
Mobility	2.1 (4.4)	4.4 (6.9)	8.8 (11.3)	<0.001
TYM score	33.0 (12.1)	28.8 (13.2)	25.7 (13.2)	<0.001
MNA	12.9 (1.5)	11.7 (2.2)	10.3 (2.9)	<0.001
CIRS-SI	2.0 (1.5)	3.5 (1.6)	4.7 (2.0)	<0.001
Number of medications	2.9 (1.8)	4.9 (2.9)	6.7 (3.1)	<0.001
Living alone (%)	7.9	29.0	31.5	<0.001
SELFY-MPI score	0.18 (0.06)	0.35 (0.03)	0.53 (0.09)	<0.001

Abbreviations: ADL: activities of daily living; IADL: instrumental activities of daily living; TYM: test your memory; MNA: mini nutritional assessment; CIRS: cumulative illness rating scale, severity index; MPI: multidimensional prognostic index.

**Table 2 vaccines-10-00555-t002:** Frequency of vaccinations in the participants included.

Type of Vaccination	Yes	No	Don’t Remember/Don’t Know
Influenza	70.5	27.9	1.5
Pneumococcus	37.6	58.0	4.4
Herpes zoster	2.5	91.2	6.3
Diphtheria-tetanus-pertussis	1.3	82.8	15.9

## Data Availability

The data presented in this study are available upon request from the corresponding author.

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
