# Peer review of "Multidimensional Frailty and Vaccinations in Older People: A Cross-Sectional Study"

_vaccines, 2022, doi:10.3390/vaccines10040555_

Round 1

Reviewer 1 Report

Q1. Lines 99-104. It is not clear how did you chose the cut-offs. Please explain

Q2. Lines 104-110. It is not clear if both questionaries were anonymous or not. If not, it would have been more correct to cross-reference the results of the cognitive tests with the Sicilian vaccination database, for example through the ID unique code. The use of a questionnaire to assess vaccination coverage, albeit a validated tool, exposes itself to bias. Please add this rating within the boundaries of your study if necessary.

Q3. Enforce conclusions, focusing on the impact of your results on public health policy.

Q4. In introduction, you should better explain the definition of "frailty older adults", reporting examples of comorbidities associated with that condition.

Author Response

Reviewer 1

Q1. Lines 99-104. It is not clear how did you chose the cut-offs. Please explain.

R: Thank you for the comment. As mentioned 0.25 and 0.38 were used for creating tertiles, whilst the prevalence of multidimensional pre-frailty and frailty was assessed using the common cut-offs for MPI, i.e., 0.33 and 0.66.

Q2. Lines 104-110. It is not clear if both questionaries were anonymous or not. If not, it would have been more correct to cross-reference the results of the cognitive tests with the Sicilian vaccination database, for example through the ID unique code. The use of a questionnaire to assess vaccination coverage, albeit a validated tool, exposes itself to bias. Please add this rating within the boundaries of your study if necessary.

R: Good point. We have added that the questionnaire is anonymous and based on self-reported information that can be biased. Therefore, in the Limitations section, we have added the following comment:

“However, we believe that for epidemiological purposes the use of self-reported tools should be encouraged, since it is important for evaluating the perceived needs and health status of older participants [38], whilst the use of a questionnaire to assess vaccination coverage can introduce a recall bias.”

Q3. Enforce conclusions, focusing on the impact of your results on public health policy.

R: Thank you for this comment. We have now added this paragraph in the conclusion section, as follows:

“Our study underlines, again, the important problem of considering frailty in terms of multidimensional approach in older people, also for suggesting vaccinations that can prevent the transition from frailty to disability.”

Q4. In introduction, you should better explain the definition of "frailty older adults", reporting examples of comorbidities associated with that condition.

R: Added.

Reviewer 2 Report

I have read with interest the manuscript entitled: Multidimensional frailty and vaccinations in older people: a cross-sectional study. This is a well-written and concise manuscript. Moreover, the investigation of vaccinations among the elderly in relation to frailty (multidimensional prognostic index) is a novelty and a significant contribution to the current body of evidence. I have some comments to make and I hope that these would be helpful for the authors.

  1. Abstract. Please provide the response rate of the study (is it 100%?).
  2. Methods. Please provide the response rate.
  3. Discussion. Apart from the association between frailty and vaccinations, the authors should expand on the descriptive characteristics of vaccination coverage against several biological hazards. They did it for influenza but there is a need to do this for other pathogens. Indeed, the literature is limited. The following reference may be of use: Papagiannis et al: Vaccination coverage of the elderly in Greece: A cross-sectional nationwide study. Can J Inf Dis Med Microbiol 2020.
  4. Discussion- limitations. The results of the present study are locally or regionally applicable due to the nature of the sample study. Thus, authors should comment on the external validity (generalisability) of their findings.

Author Response

I have read with interest the manuscript entitled: Multidimensional frailty and vaccinations in older people: a cross-sectional study. This is a well-written and concise manuscript. Moreover, the investigation of vaccinations among the elderly in relation to frailty (multidimensional prognostic index) is a novelty and a significant contribution to the current body of evidence. I have some comments to make and I hope that these would be helpful for the authors.

  1. Abstract. Please provide the response rate of the study (is it 100%?).

R: Added.

  1. Methods. Please provide the response rate.

R: Thank you. We added that the response rate was 319 of over 500 participants (63.8%).

  1. Apart from the association between frailty and vaccinations, the authors should expand on the descriptive characteristics of vaccination coverage against several biological hazards. They did it for influenza but there is a need to do this for other pathogens. Indeed, the literature is limited. The following reference may be of use: Papagiannis et al: Vaccination coverage of the elderly in Greece: A cross-sectional nationwide study. Can J Inf Dis Med Microbiol 2020.

R: We would like to sincerely thank the Reviewer for suggesting such a nice contribution. We have added this important paper in the Discussion and discussed the relevance for other pathogens, as suggested.  

  1. Discussion- limitations. The results of the present study are locally or regionally applicable due to the nature of the sample study. Thus, authors should comment on the external validity (generalisability) of their findings.

R: We fully agree with this comment. We have now added this, as potential limitation of the work, as follows:

“Finally, the results of the present study are locally or regionally applicable; therefore, the external validity (generalizability) of these findings must be verified in larger studies conducted in other populations.”

Reviewer 3 Report

The paper presents the results of study assessing the association between vaccination coverage and frailty in a sample of 319 individuals aged 60 years and above accessing an outpatient clinic in Italy.

The paper has the following problems:

  1. Line 32-33. “Frailty and vaccination status … appears to be associated”. Nevertheless, the study did not assess the association between frailty (defined as risk score >0.66) and vaccination coverage. Only pneumococcal vaccine coverage was significantly associated with MPI groups, but it was similar in MPI2 and MPI3.  
  2. Lines 58-62. The key objective of the study seems to assess the vaccination coverage in a cohort of elderly people, but the text of the paper is mainly focused on the association between MPI groups and vaccination coverage. The research question must be indicated in a clear way.
  3. Lines 66-68. The sampling method is not explained in sufficient detail.
  4. Lines 69-70. “The caregivers were allowed to assist the participants in filling out the questionnaire”. What information had the caregivers about the vaccines received by participants?
  5. Lines 73-96. What is the reason of assessing frailty using such a complex method including 8 different tools? The Methods section must explain what methods can be used to assess frailty and why in this study a method for assessing mortality risk was used to asses frailty.
  6. Lines 103-103. “The median time to complete the MPI is about 16 minutes”. This time seems difficult.
  7. Lines 104-110. It is necessary to explain current vaccination recommendations in individuals aged 60 years or above in Italy, including references.
  8. Lines 112-120. The statistical analysis section is very poor. If the objective of the study was to assess the association between frailty and vaccination coverage, it is necessary to explain the research strategy, the study variables, what was assessed and compared, statistical tests used and p values considered.
  9. Line 122. The sample size calculation is not presented. Why 500 individuals were invited to the study?
  10. Lines 126-131. The results presented on table 1 are not explained in sufficient detail.
  11. Line 205. The reasons for detecting frailty using a method for determining prognostic should be discussed in the paper. Why a less complex method for detecting frailty was not used in the study?
  12. Line 208. The prevalence of frailty observed in the study (3.4%) was lower than the 5% indicated in this sentence. This difference should be discussed.
  13. Lines 241-246. An important limitation of the study is that generalization of results from a sample to the population must be based on the representativeness of the sample studied. The sampling method could not ensure the representativeness of the sample of individuals studied.
  14. Lines 243-244. “Since self-reported information regarding frailty and vaccination status were used, a recall bias could be present”. Other potential bias could be gebnerated by caregiver assistance, as mentioned on lines 69-70. How potential bias were detected and controlled in the study?
  15. Lines 248-252. The conclusion about the “high prevalence of frailty” and that “frailty and vaccination status appear to be associated” are not supported by the results obtained in the study. A risk score >0.38 (MPI3) is not equivalent to frailty. Vaccination coverages were similar in different MPI groups except for pneumococcal vaccine, which was similar in MPI2 and MPI3 groups.

Author Response

The paper presents the results of study assessing the association between vaccination coverage and frailty in a sample of 319 individuals aged 60 years and above accessing an outpatient clinic in Italy.

The paper has the following problems:

  1. Line 32-33. “Frailty and vaccination status … appears to be associated”. Nevertheless, the study did not assess the association between frailty (defined as risk score >0.66) and vaccination coverage. Only pneumococcal vaccine coverage was significantly associated with MPI groups, but it was similar in MPI2 and MPI3. 

R: We agree with your pertinent comment. We have now remodeled the conclusion of our Abstract, as follows:

“Higher SELFY-MPI scores and vaccination status, particularly anti-pneumococcus, appear to be associated, but future studies are urgently needed for confirming that frailty is associated with vaccination status in older people.”

  1. Lines 58-62. The key objective of the study seems to assess the vaccination coverage in a cohort of elderly people, but the text of the paper is mainly focused on the association between MPI groups and vaccination coverage. The research question must be indicated in a clear way.

R: We sincerely thank the Reviewer for this comment. We have changed the aim of the study, accordingly.

  1. Lines 66-68. The sampling method is not explained in sufficient detail.

R: Thank you for the question. We have now added this detail in the Methods section:

“Possible eligible participants were contacted during the administration of SarsCov2 vaccination by physicians in training in geriatric medicine of the University of Palermo that explained the main aims of the work to the possible participants and their caregivers.”

  1. Lines 69-70. “The caregivers were allowed to assist the participants in filling out the questionnaire”. What information had the caregivers about the vaccines received by participants?

R: With this sentence, we mean that the caregivers can help the older participants with the SELFY-MPI and with the questions regarding vaccination status. We have now better explained this point.

  1. Lines 73-96. What is the reason of assessing frailty using such a complex method including 8 different tools? The Methods section must explain what methods can be used to assess frailty and why in this study a method for assessing mortality risk was used to asses frailty.

R: Good point. We have now added this explanation in the Methods section, for better explaining the concept that MPI can be used for assessing multidimensional frailty:

“MPI is commonly used as tool for evaluating the presence of frailty, assessed through a comprehensive geriatric assessment, i.e., multidimensional frailty.[21]”

  1. Lines 103-103. “The median time to complete the MPI is about 16 minutes”. This time seems difficult.

R: We understand that it may seem quite long, particularly if referred to an older population. However, this time is needed in order to complete a self-assessment of frailty using a comprehensive geriatric assessment tool. Nevertheless, a longer time is generally necessary to complete the battery of tests included in the usual comprehensive geriatric assessment.

  1. Lines 104-110. It is necessary to explain current vaccination recommendations in individuals aged 60 years or above in Italy, including references.

R: We added a recent reference (22) supporting the current vaccination recommendations in older people, as suggested.

  1. Lines 112-120. The statistical analysis section is very poor. If the objective of the study was to assess the association between frailty and vaccination coverage, it is necessary to explain the research strategy, the study variables, what was assessed and compared, statistical tests used and p values considered.

R: Thank you for your considerations. We have added some details in the Statistical Section analysis, as suggested.

  1. Line 122. The sample size calculation is not presented. Why 500 individuals were invited to the study?

R: Good point. We have added, as follows, as the sample size was calculated:

“Sample size was calculated according to a work investigating the prevalence of multidimensional frailty with the SELFY-MPI among general practitioners in Italy.[23] In this regard, hypothesizing a prevalence of multidimensional frailty of 3.99%, a type-I error of 5% and a power of 80%, 246 participants should be enrolled.”

  1. Lines 126-131. The results presented on table 1 are not explained in sufficient detail.

R: Added.

  1. Line 205. The reasons for detecting frailty using a method for determining prognostic should be discussed in the paper. Why a less complex method for detecting frailty was not used in the study?

R: Good point. In the Discussion section we have better reported why frailty was assessed using a multidimensional tool, as follows:

“Increasing literature is reporting that the presence of frailty should be assessed using comprehensive geriatric assessment and so with a multidimensional tool. [21,40]”

  1. Line 208. The prevalence of frailty observed in the study (3.4%) was lower than the 5% indicated in this sentence. This difference should be discussed.

R: Good point. We added the following comment in our Discussion section:

“In our study, we found a slightly lower prevalence of frailty and pre-frailty according to a multidimensional approach, probably because these older people were able to access the services for receiving vaccination against SarsCov2; therefore, bedridden and more disabled older participants were not included.”

  1. Lines 241-246. An important limitation of the study is that generalization of results from a sample to the population must be based on the representativeness of the sample studied. The sampling method could not ensure the representativeness of the sample of individuals studied.

R: We fully agree with this consideration, as correctly observed by another Reviewer. Therefore, we added this as limitation of our study, as follows:

“Finally, the results of the present study are locally or regionally applicable; therefore, the external validity (generalizability) of these findings must be verified in larger studies conducted in other populations.”

  1. Lines 243-244. “Since self-reported information regarding frailty and vaccination status were used, a recall bias could be present”. Other potential bias could be generated by caregiver assistance, as mentioned on lines 69-70. How potential bias were detected and controlled in the study?

R: Thank you so much for this comment. We have reported in the Results section that a caregiver was present for several participants (64.9% of the participants). Therefore, we commented this finding, as follows:

“Second, since self-reported information regarding frailty and vaccination status were used, a recall bias could be present as well as the caregiver assistance that was necessary for several participants.”

  1. Lines 248-252. The conclusion about the “high prevalence of frailty” and that “frailty and vaccination status appear to be associated” are not supported by the results obtained in the study. A risk score >0.38 (MPI3) is not equivalent to frailty. Vaccination coverages were similar in different MPI groups except for pneumococcal vaccine, which was similar in MPI2 and MPI3 groups.

R: We sincerely thank the Reviewer for this comment. We have now modified our Conclusion section, as made in the Abstract, as follows, for better reflecting our findings:

“Our study suggests that higher SELFY-MPI scores, indicating more propensity to a multi-dimensional frailty, and vaccination status, particularly anti-pneumococcus, appear to be associated.”

Round 2

Reviewer 2 Report

I have read the revised version of the above mentioned manuscript.

I believe that authors have effectively responded to all comments made be the reviewers. On the bases of the above, I believe that the manuscript can be published in Vaccines.

Reviewer 3 Report

The quality of the revised vesrion of the paper has increased. The authors have provided necessary details of the methodology, and have differentiated frailty from MPI score groups.